# Long-term intensive golf training induces reconfiguration of brain structural covariance networks

Zonghan Lei[1], Yaoqi Hou[2], Xiangqin Song[2]*

**1** Business School, University of New South Wales, UNSW Sydney High St Kensington, Sydney, New South Wales, Australia, **2** College of P.E. and Sports, Beijing Normal University, Beijing, China

* u252016415@163.com

## Abstract

Long-term motor training is thought to reshape brain organization, yet how golf expertise influences large-scale brain networks remains unclear. Using T1-weighted MRI and an individualized structural covariance network (SCN) approach, we compared 20 expert golfers, 20 novice golfers, and 20 non-golfer controls. Experts showed higher global clustering coefficient and local efficiency than novices, indicating enhanced modular processing. At the nodal level, experts exhibited increased clustering in regions supporting visual–sensorimotor integration (e.g., right supramarginal gyrus, Heschl's gyrus, and left middle temporal pole), alongside reduced global efficiency in the left calcarine cortex and altered path length in the right cerebellum. Importantly, the clustering coefficient mediated the association between training duration and stroke accuracy. These cross-sectional findings suggest that extensive golf training is linked to a brain network reconfiguration that favors local specialization over global integration—potentially supporting the refined sensorimotor control required in elite performance. This study advances understanding of experience-dependent neuroplasticity by integrating individualized network analysis with behavioral outcomes in motor expertise.

## 1. Introduction

Motor skill learning, characterized by progressive improvements in movement speed and accuracy through repeated practice, is fundamental to athletic training and performance [1,2]. This process reflects the brain's remarkable capacity for plasticity, whereby repeated motor experience induces structural and functional adaptations that support expertise [3]. Athletes engaged in long-term intensive training from an early age often develop superior motor skills—such as coordination, agility, and endurance—that are underpinned by lasting changes in brain structure, particularly within regions responsible for sensory-motor integration and cognitive control [4].

**Data availability statement:** The analysis code and processed data are publicly available at: https://github.com/russellei/Long-Term-Intensive-Golf-Training-Induces-Reconfiguration-of-brain-structural-covariance-networks.

**Funding:** The author(s) received no specific funding for this work.

**Competing interests:** The authors have declared that no competing interests exist. We confirm that there are no financial, personal, or professional interests that could be perceived to influence the objectivity of this research. All sources of funding supporting the work described in this manuscript have been disclosed in the Funding Information section.

Early research on brain plasticity associated with motor expertise has primarily examined isolated brain regions, frequently reporting changes in gray matter volume. For example, structural enlargements in the hippocampus have been documented in London taxi drivers, reflecting their extensive navigational experience [5–7]. Similarly, basketball players and soccer players exhibit greater gray matter volumes in motor cortices and the cerebellum compared to novices [8]. Conversely, reductions in gray matter volume have been observed in the sensorimotor cortex of ballet dancers [9–11] and in the visual cortex of elite gymnasts [12,13], which may represent neural refinement and efficiency in motor processing. Collectively, these findings highlight that long-term training can both increase and decrease gray matter volume depending on task demands and expertise type.

Despite these valuable insights, most prior studies have relied on univariate approaches such as voxel-based morphometry (VBM), which focus on region-specific morphological changes while overlooking the brain's complex, interconnected nature. Advances in network neuroscience emphasize that brain regions operate as integrated systems rather than in isolation [14,15]. Thus, understanding how long-term intensive training reconfigures structural covariance networks (SCNs) may provide a more comprehensive account of the neural mechanisms that support expert performance [16,17].

To address this gap, the present study applies a network-based approach to investigate how long-term golf training influences brain structural covariance networks. By constructing individualized SCNs, this research moves beyond traditional group-level analyses and enables the characterization of network reorganization at both global and modular levels. This methodological innovation allows us to explore how expert training reshapes the structural architecture of the brain, bridging the gap between local morphological changes and system-level adaptations. Through this framework, we aim to provide new insights into the neural mechanisms of motor expertise, with implications for both athletic training and rehabilitation.

## 2. Subject and method

### 2.1. Subject

A total of 60 participants were recruited for this study and divided into three groups: the expert golfer group (Pro), the novice golfer group (New), and a control group. The Pro group comprised 20 highly skilled and experienced golfers, with an average score of 72.05±4.3 strokes per 18 holes. These individuals trained at least six times per week, with each session lasting a minimum of two hours, and had accumulated no less than 10 years of training experience. The New group included 20 participants with limited golf experience, averaging 86.15±5.63 strokes per 18 holes. They had fewer than three years of training experience and practiced approximately three times per week, with each session lasting around 1.5 hours. Participant expertise classification was based on criteria adapted from previous studies [18].

The control group consisted of 20 individuals with no formal golf training. All three groups were matched in terms of age, gender, and educational background, and no statistically significant differences were observed among them in age or body mass

index (BMI) (see Table 1). All participants were right-handed, had normal or corrected-to-normal vision, reported no history of neurological or psychiatric disorders, and met the safety requirements for MRI scanning.

Participant recruitment occurred between 01/05/2023 and 15/12/2023. The study was approved by the Ethics Committee of the College of Physical Education and Sport Sciences, Beijing Normal University (Project Number: 133325607). Written informed consent was obtained from all participants. All research materials and findings were strictly used for scientific purposes, with no conflicts of interest. The experimental design and protocol were reviewed and deemed to adequately consider safety and fairness by the Ethics Committee. Participation was voluntary, and participants' rights and privacy were fully protected.

## 2.2. Image acquisition and preprocessing

All the subjects in this study were scanned with a Siemens 3.0T MRI scanner (Trio Tim, Erlangen, Germany) using an eight-channel head coil, including high-resolution T1 weighted imaging (T1WI). Te parameters were as follows: T1WI using a Magnetization Prepared Rapid Gradient Echo (MPRAGE) sequence, TR/TE = 2250/2.6 ms, slice thickness = 1 mm, fip angle = 9°, FOV = 256 × 256 mm2, and matrix size = 256 × 256.

The present study utilized high-resolution T1-weighted imaging data, which underwent preprocessing with the CAT12 toolbox of the SPM12 software package in order to eliminate motion and other artifacts. CAT12 is considered to be more accurate than VBM8 for analysis of volume changes. All participants' images were segmented into white matter, gray matter, and cerebrospinal fluid. Subsequently, the images were normalized to the Montreal Neurological Institute (MNI) standard space, and modulated using the Difeomorphic Anatomical Registration Trough Exponential Lie Algebra toolbox. Finally, the rest gray matter images were smoothed with a Gaussian kernel of 8 mm full-width at half maximum.

## 2.3. Network constructing

All structural analyses were based on the Automated Anatomical Labeling (AAL) atlas, which divides the brain into 116 regions of interest (ROIs). Gray matter volumes (GMVs) were extracted from each ROI and served as the basis for network construction. As the traditional group-level structural covariance network(SCN) lost the individual network information, here we adapted a recently established approach based on the inter-regional effect size difference (ESD) to obtain individual SCN for our expert-novice study [19,20]. To retain individual network information, we used an approach based on inter-regional effect size difference (ESD) to obtain individual SCN for our case-control study. The main steps to construct individual SCN for each participant within each cohort are as follows (see the Fig 1): (1) Regional gray matter volume(GMV) for each region of interest were adjusted for covariates (e.g., age, gender, and total intracranial volume), and the resulting regional GMV residuals were extracted. (2) A group-based SCN ($SCN_{conotrol}$) was created across the entire control group by calculating the Pearson correlation coefficient between the regional GMV residuals for each pair of brain regions. (3) We computed the mean ($M_{control}$) and standard deviation ($SD_{control}$) values of each brain region from the control group. (4) The individual weight matrix (W) was derived from the inter-regional ESD between a single subject and the average of the control group. (5) The final individual correlation coefficient matrix for a single subject was then calculated

**Table 1. Demographic and information of subjects.**

| Group | Pro(n = 20) | New(n = 20) | Control(n = 20) |
|---|---|---|---|
| Age | 27.4 ± 3.1 | 27.1 ± 2.8 | 27.6 ± 2.2 |
| BMI | 20.5 ± 1.6 | 21.3 ± 1.8 | 21.9 ± 1.2 |
| Strokes per 18 holes | 72.05 ± 4.3 | 86.15 ± 5.63 | |
| Training duration | 12.4 ± 1.2 | 1.8 ± 1.1 | |

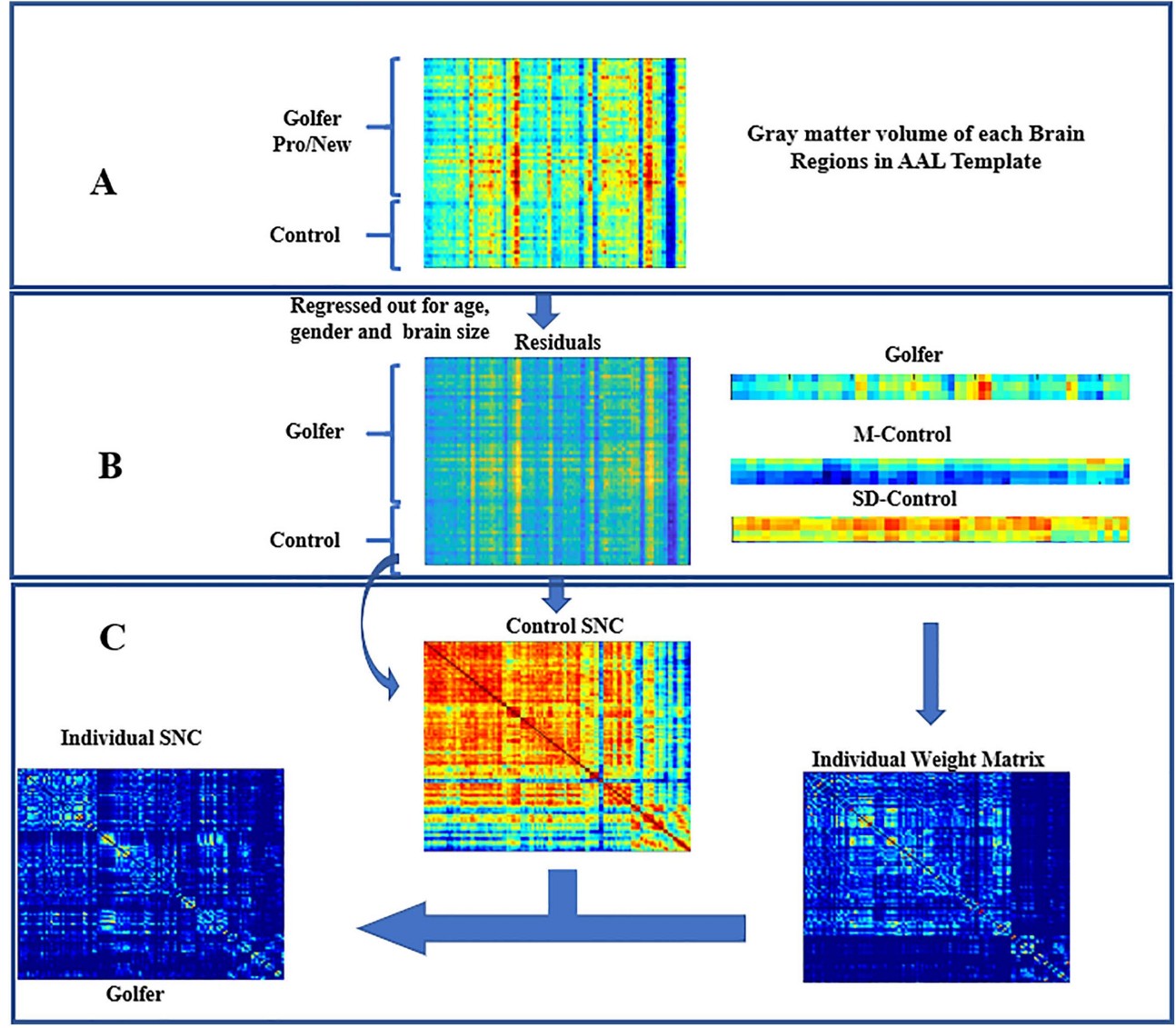

**Fig 1. Schematic workflow of the contribution of structural covariance network.** Panel A shows the initial step of extracting regional gray matter volume (GMV) data from T1-weighted MRI scans using the Automated Anatomical Labeling (AAL) atlas, illustrated here for two example participants (Golfer Pro/New and Control). Panel B depicts the subsequent computational pipeline: GMV values are first adjusted for covariates (age, gender, and total brain volume) to obtain residuals, and then a group-level SCN (Control SCN) is constructed from healthy controls (HCs). Panel C illustrates the final steps: an individual weight matrix is computed based on the deviation of a subject's GMV residuals from the control group mean (M-Control) and standard deviation (SD-Control); the individual SCN for each participant (e.g., Golfer) is then derived by combining the group-level SCN with the individual weight matrix. M, mean; SD, standard deviation; SCN, structural covariance network.

by element-wise multiplication between W and SCN$_{conotrol}$. Using this method, we obtained a $116 \times 116$ connection matrix for each subject in the golf expert and novice groups.

## 2.4. Network analysis

A range of sparsity levels (K = 0.14–0.50, step size = 0.01) was applied to binarize individual structural covariance networks (SCNs), which were then used for group comparisons of both global and nodal network properties. The sparsity

range was determined based on established criteria from previous studies [21,22], ensuring that: (1) over 90% of nodes remained interconnected within each network, and (2) more than 95% of individual SCN graphs exhibited a small-world index greater than 1, indicating prominent small-world characteristics. At each sparsity level, four global network metrics were calculated: global efficiency (Eg), local efficiency (Eloc), average path length (Lp), and clustering coefficient (Cp). In addition, two regional (nodal) topological measures—nodal degree and betweenness centrality—were also computed for each node using the Brain Connectivity Toolbox [23]. The detailed computational procedures are presented in Table 2. To capture the overall performance of each topological property across the full range of sparsity thresholds, the area under the curve (AUC) was calculated for every metric and subsequently used in the statistical analyses.

## 2.5. Mediation analysis

To further explore the interrelationships among training duration, brain structural covariance network properties, and pole performance, we conducted a mediation analysis. Drawing upon findings from Li et al., we hypothesized that neuroplasticity—reflected in changes in brain network organization—plays a mediating role in linking training duration to specific behavioral outcomes.

First, we performed correlation analyses to identify which network metrics showed significant associations with both training duration and pole performance. These network parameters were then selected as potential mediators in the subsequent mediation models. Specifically, traditional mediation involved estimating paths, including direct (path c) and indirect effects (path c') of the independent variable X on the dependent variable Y (Fig 2). The direct effect of X on Y after considering the mediator M (path c') and the indirect effect through M (i.e., the product of X→M and M→Y paths). Mediation was assessed using nonparametric bootstrapping (5000 samples) to generate bias-corrected 95% confidence intervals (CIs) for the a×b sampling distribution. A CI excluding zero indicates a significant deviation from zero for the indirect effect (path a×b, P<.05).

## 2.6. Statistical analysis

To examine group differences in network metrics, we employed a permutation-based nonparametric approach. This method does not assume a specific distribution for the data, thus providing a robust and distribution-free assessment of statistical significance. Specifically, for each network metric, the observed mean difference between groups was calculated and compared against a null distribution generated by randomly permuting group labels 5000 times. The p-value for each comparison was defined as the proportion of permuted differences that were equal to or greater than the observed difference. To control for multiple comparisons across all network metrics and brain regions, the false discovery rate (FDR) correction was applied. Additionally, Cohen's d effect sizes were calculated alongside p-values to quantify the magnitude

**Table 2. The description of global network parameters.**

| Network parameters | Descriptions |
|---|---|
| Clustering coefficient | It indicates the extent of local interconnectivity or cliquishness in a network. |
| Average path length | It measures a harmonic mean length between pairs and quantifies the ability for information propagation in parallel. |
| Global efficiency | It is the inverse of the harmonic mean of the characteristic path length between each pair of nodes within the network. $E_{glob}$ measures the global efficiency of parallel information transfer in the network. |
| Local efficiency | It is defined as the average of the values of local efficiency across all nodes. |

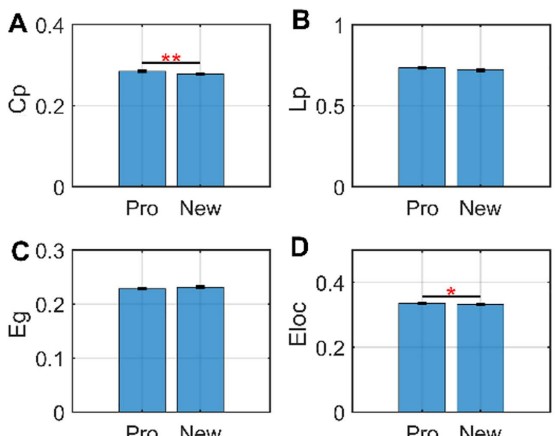

**Fig 2. Standard 3-variable path model of mediation analysis.**

of between-group differences and to facilitate a comprehensive interpretation of both statistical and practical significance. For correlation analyses, we first assessed the normality of all variables using the Shapiro–Wilk test. Variables that met the assumption of normality ($p > 0.05$) were analyzed using Pearson's correlation, whereas variables that violated normality were analyzed using Spearman's rank correlation. In the present study, all variables included in the correlation analyses conformed to a normal distribution; therefore, Pearson's correlation was used throughout.

## 3. Results

### 3.1. Alterations in global network topologies in golf experts

The comparison of global network topologies between golf experts and novices revealed significant differences. Specifically, golf experts exhibited a higher Cp and greater Eloc in their brain structural covariance networks compared to novices (Fig 3, Table 3).

**Fig 3. Comparison of global network topological properties between golf experts and novices.** "*" indicates significant difference at $p < 0.05$, and "**" indicates $p < 0.01$, both after False Discovery Rate (FDR) correction.

**Table 3. Comparison of global network topological properties between golf experts and novices.**

|  | Pro | New | P-FDR | Cohen'd |
|---|---|---|---|---|
| Cp | 0.2850 ± 0.0061 | 0.2786 ± 0.0057 | 0.012 | 1.02 |
| Lp | 0.7336 ± 0.019 | 0.72 ± 0.0221 | 0.631 | 0.63 |
| Eloc | 0.3363 ± 0.0046 | 0.3322 ± 0.0047 | 0.004 | 0.69 |
| Eg | 0.2282 ± 0.0049 | 0.2313 ± 0.0057 | 0.724 | −0.53 |

## 3.2. Alteration in regional network topologies in golf experts

At the nodal level, distinct differences were observed between experts and novices (Fig 4, Table 4). Golf experts demonstrated a higher clustering coefficient in the right supramarginal gyrus, Heschl's gyrus, and the left middle temporal pole. Additionally, greater local efficiency was noted in the left middle temporal pole. Conversely, experts exhibited lower global efficiency in the left calcarine cortex and the middle temporal pole, along with a shorter path length in the right cerebellum-10 compared to novices.

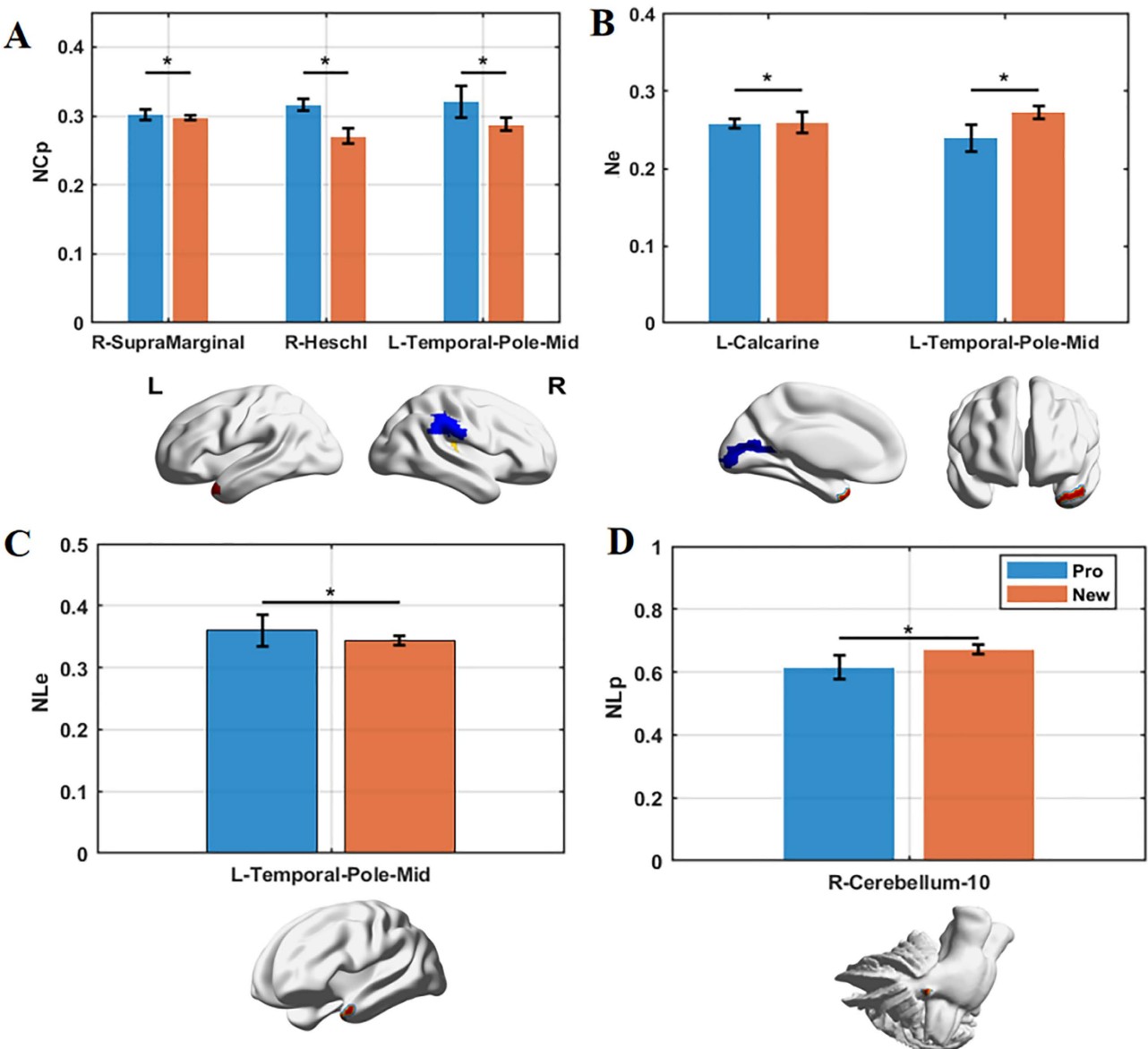

**Fig 4. Group differences in nodal-level network topological properties between golf experts and novices.** "*" indicates significant differences at p<0.05 after FDR correction. (Cp: clustering coefficient, Lp: path length, Le: local efficiency, Ne: nodal efficiency).

**Table 4. Comparison of nodal network topological properties between golf experts and novices.**

|  | Brain Regions | Pro | New | P-FDR | Cohen'd |
|---|---|---|---|---|---|
| NCp | R-SupraMarginal | 0.3017±0.0073 | 0.297±0.0038 | 0.013 | 0.65 |
|  | R-Heschl | 0.316±0.0082 | 0.271±0.0112 | 0.014 | 0.72 |
|  | L-Temporal-Pole-Mid | 0.321±0.0221 | 0.288±0.0087 | 0.012 | 0.68 |
| NLe | L-Temporal-Pole-Mid | 0.3601±0.025 | 0.3439±0.008 | 0.021 | 0.58 |
| Ne | L-Calcarine | 0.2587±0.0061 | 0.2596±0.013 | 0.027 | 0.52 |
|  | L-Temporal-Pole-Mid | 0.240±0.0178 | 0.273±0.0084 | 0.018 | 0.61 |
| NLp | R-Cerebellum Region 10 | 0.615±0.0373 | 0.673±0.0152 | 0.009 | 0.75 |

### 3.3. Mediated analysis results

Among network metrics showing significant group differences, global clustering coefficient (Cp) and nodal local efficiency of the left middle temporal pole (NLe) were both significantly and negatively correlated with stroke performance (Fig 5A–B). This indicates that higher network modularity and regional efficiency are associated with better golf performance (fewer strokes per round). Additionally, training duration was positively correlated with Cp (Fig 5C), suggesting that longer practice is linked to greater structural network segregation.

Mediation analysis confirmed that Cp significantly mediates the relationship between training duration and stroke performance (Fig 5D). Training duration positively predicted Cp ($\beta = 0.33$, $p < 0.001$), which in turn negatively predicted stroke count ($\beta = -0.31$, $p < 0.001$). The indirect effect was statistically significant ($\beta = -0.1$, 95% CI [−0.37, −0.09]), while the direct effect of training duration on stroke performance became non-significant after controlling for Cp ($\beta = -0.22$, $p = 0.15$).

## 4. Discussion

This study reveals that long-term golf training is associated with a reconfigured structural covariance network architecture characterized by enhanced modularity and region-specific neuroplasticity. At the global level, expert golfers exhibited higher clustering coefficient (Cp) and local efficiency (Eloc) compared to novices, indicating a shift toward more specialized, locally integrated processing units. This pattern aligns with theoretical models suggesting that expertise optimizes information processing by segregating domain-relevant functions into dedicated subsystems while minimizing costly long-range communication [24,25].

Notably, these global changes co-occurred with nodal-level alterations in regions critical for sensorimotor integration and visuospatial control. For instance, increased clustering in the right supramarginal gyrus—a hub for body schema updating and visuomotor coordination [26,27]—may support the precise postural adjustments required during a golf swing. Similarly, structural reorganization in Heschl's gyrus and the left middle temporal pole could reflect enhanced auditory-spatial processing and high-level semantic integration of environmental cues. The temporal pole is primarily functionally connected with higher-order visual cortical regions [28], and its distinct connectivity with the vestibular nuclei suggests a potential role in modulating eye movement control and sensorimotor coordination [29]. This finding is consistent with prior work showing greater gray matter volume in the left middle temporal pole of golf experts, which was positively correlated with stroke performance [18].

Conversely, reduced global efficiency in the left calcarine cortex and altered path length in the right Cerebellum Region 10 suggest a strategic trade-off: the expert brain may prioritize local computational precision over broad integrative capacity, a hallmark of skill automatization [30]. This reduction in global efficiency within the left calcarine cortex—the primary visual area—may appear counterintuitive at first glance, as expertise is often associated with enhanced neural resources. However, from a neural efficiency perspective, such a decrease can be interpreted as a functional pruning process, wherein long-term training refines visual processing by suppressing task-irrelevant or redundant connections while preserving and strengthening behaviorally relevant pathways [24,31]. In the context of golf, athletes must maintain

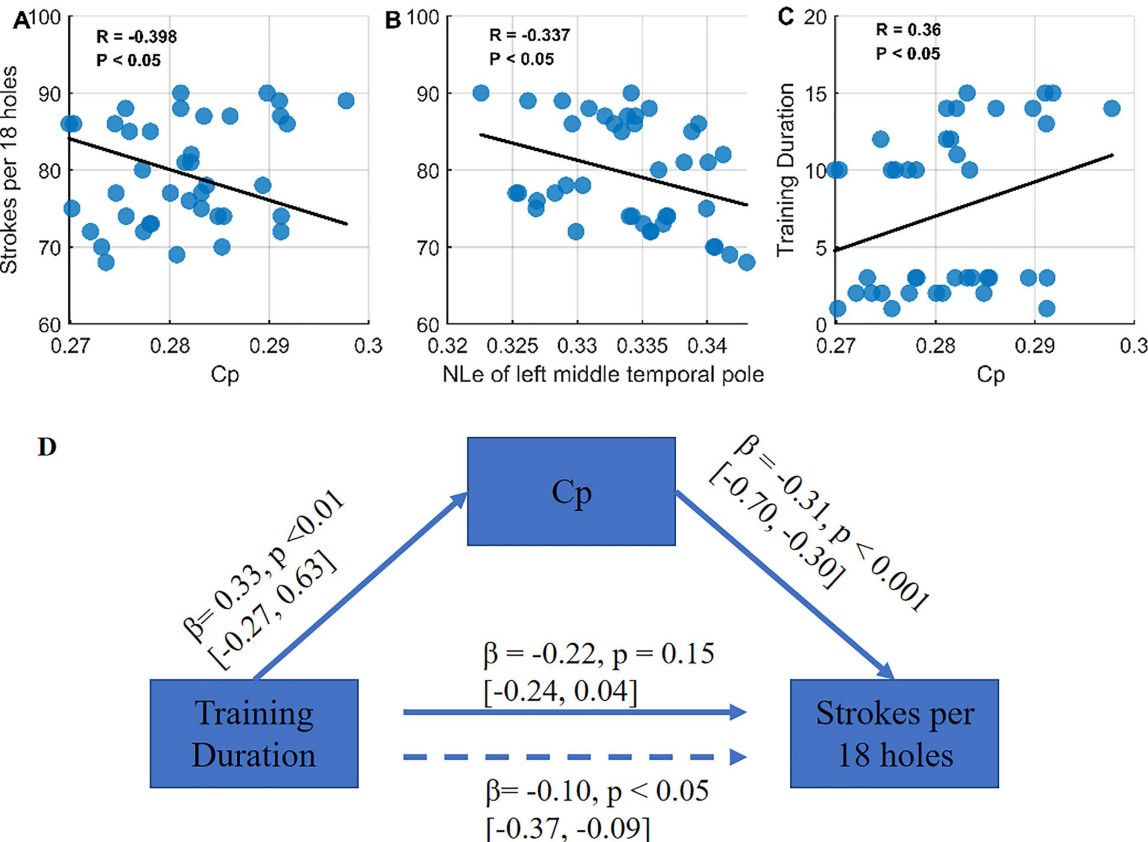

**Fig 5. The relationship among structural covariance network topologies, strokes per 18 holes and training duration.** Subplots A–C show the results of the correlational analyses. Subplot D presents the results of a standard three-variable mediation model, indicating that Cp mediates the relationship between training duration and stroke performance. The dashed line represents the indirect effect, while the solid line represents the direct effect.

stable visual fixation and ignore peripheral distractors during the swing and putting phases. A more locally specialized, less globally integrated visual network may therefore facilitate focused attention and reduced perceptual noise, ultimately supporting higher consistency in stroke execution. This interpretation aligns with findings in other expert populations, such as gymnasts and shooters, who exhibit reduced cortical volume or activity in visual regions alongside superior visuomotor performance [17,32]. Thus, rather than reflecting neural degradation, the observed decrease in nodal global efficiency likely represents an adaptive reallocation of neurocomputational resources toward local, task-specific processing. This aligns with the concept of "neural efficiency" observed in other domains of expertise, where sustained, domain-specific training drives structural plasticity in functionally relevant areas [27,33–35].

The mediation analysis further demonstrates that the clustering coefficient (Cp) significantly mediated the relationship between training duration and golf performance (i.e., average strokes per round). This suggests that structural network reconfiguration, particularly at the local level, serves as a neural substrate underlying behavioral improvements. Importantly, this mediation effect highlights the potential utility of Cp as a neurobiological marker for tracking expertise development—a perspective that could inform future longitudinal or intervention-based studies on skill acquisition.

Our network-based approach offers a novel perspective that extends beyond traditional studies focusing on isolated brain regions. Rather than examining localized volume changes alone, we emphasize how extensive practice reshapes interregional structural covariance across the entire brain network. This systems-level view supports the hypothesis that

expertise is not merely a function of isolated neural hypertrophy but rather an emergent property of optimized, coordinated interactions among distributed brain regions [36]. Such structural adaptations likely underpin the superior motor control, sensory integration, and visuospatial processing abilities characteristic of expert performers [35].

Our findings are consistent with previous research on brain functional networks in athletes, which have reported similar modularity and efficiency patterns in individuals such as soccer players and endurance athletes [15,16]. Extending these observations to structural covariance networks, our results reinforce the idea that both functional and structural architectures undergo parallel reorganization in response to intensive skill acquisition. Although structural connectivity provides the anatomical substrate for functional dynamics, the precise mechanisms linking structure to function remain incompletely understood [37–39]. Future multimodal neuroimaging studies integrating structural, functional, and behavioral data will be essential for unraveling this complex relationship.

The increased modularity observed in experts' brain networks may reflect a shift toward more parallelized processing, where distinct cognitive functions—such as visuospatial attention, motor planning, and sensory integration—are supported by relatively independent neural modules [31,40]. This organizational strategy could facilitate faster and more efficient information processing, reduce cross-talk between subsystems, and enhance overall task performance [41,42]. Such an architecture is particularly advantageous in golf, a sport that demands high levels of focused attention, precise motor control, and consistent performance under varying environmental and psychological conditions.

## 5. Limitations

Despite the novel insights of this study, several limitations should be noted. First, its cross-sectional design precludes causal inferences about the relationship between long-term training and brain network reorganization; longitudinal research is needed to clarify causality and the temporal dynamics of these changes. Second, our reliance on structural MRI limits interpretation, as structural covariance networks cannot fully capture the functional dynamics of expertise. Combining structural, functional, and diffusion MRI would provide a more comprehensive picture of how training reshapes the brain. Finally, unmeasured factors such as lifestyle, fitness, stress, or personality may also contribute to group differences. Although groups were matched on age, sex, and education, future studies should account for these potential confounders.

## 6. Conclusion

This study demonstrates that long-term golf training is associated with reorganized structural covariance networks, characterized by enhanced global modularity and local efficiency. Key sensorimotor and cognitive regions showed nodal-level changes, reflecting expertise-related neuroplasticity. Mediation analysis further revealed that local network properties, such as the clustering coefficient, bridge training duration and performance improvement. These findings suggest that prolonged motor training shapes brain architecture in ways that support superior skill execution and behavioral adaptation—not through isolated regional changes, but via system-wide network reconfiguration.

## Acknowledgments

We gratefully acknowledge all participants for their time and commitment to this study. We also thank the staff at the Imaging Center of Beijing Normal University for their technical support during MRI data acquisition.

## Author contributions

**Conceptualization:** Yaoqi Hou, Xiangqin Song.

**Data curation:** Xiangqin Song.

**Formal analysis:** Zonghan Lei.

**Investigation:** Yaoqi Hou.

**Methodology:** Zonghan Lei.

**Supervision:** Yaoqi Hou.

**Writing – original draft:** Zonghan Lei.

**Writing – review & editing:** Xiangqin Song.

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
