## [Decision Letter · Decision Letter 0]

1 Oct 2025

Dear Dr. Qiangqin Song,

Thank you for submitting your manuscript to PLOS ONE. After careful consideration, we feel that it has merit but does not fully meet PLOS ONE’s publication criteria as it currently stands. Therefore, we invite you to submit a revised version of the manuscript that addresses the points raised during the review process.

Please submit your revised manuscript by Nov 15 2025 11:59PM. If you will need more time than this to complete your revisions, please reply to this message or contact the journal office at plosone@plos.org . A rebuttal letter that responds to each point raised by the academic editor and reviewer(s). You should upload this letter as a separate file labeled 'Response to Reviewers'.A marked-up copy of your manuscript that highlights changes made to the original version. You should upload this as a separate file labeled 'Revised Manuscript with Track Changes'.An unmarked version of your revised paper without tracked changes. You should upload this as a separate file labeled 'Manuscript'.

We look forward to receiving your revised manuscript.

Kind regards,

Cory James Coehoorn

Academic Editor

PLOS ONE

Journal Requirements:

Reviewers' comments:

Reviewer's Responses to Questions

**Comments to the Author**

1. Is the manuscript technically sound, and do the data support the conclusions?

Reviewer #1: Partly

Reviewer #2: Partly

2. Has the statistical analysis been performed appropriately and rigorously?

Reviewer #1: N/A

Reviewer #2: Yes

3. Have the authors made all data underlying the findings in their manuscript fully available?

Reviewer #1: Yes

Reviewer #2: No

4. Is the manuscript presented in an intelligible fashion and written in standard English?

Reviewer #1: Yes

Reviewer #2: Yes

Reviewer #1: The abstract should be shortened and written more concisely to highlight the main findings and contributions without excessive technical detail. The description of the methods, particularly the use of inter-regional effect size differences (ESD), could be simplified for broader accessibility. In addition, the number of participants must be consistent across the abstract and the Methods section. The conclusion should clearly emphasize the novelty of the study and its implications for understanding motor expertise.

The introduction currently reviews the literature extensively but does not sufficiently emphasize the research gap. It should explicitly state the limitations of prior studies focusing on isolated brain regions and univariate analyses, and then highlight how a network-based approach provides a novel contribution. There is also an inconsistency where “soccer experts” appears instead of “golf experts,” which must be corrected throughout. The section would be stronger if it followed a clearer structure: (1) motor learning and brain plasticity, (2) evidence from previous athlete studies, (3) limitations of regional approaches, and (4) the novelty and aims of the present study.

The Results section contains a labeling error, referring to “soccer experts” instead of “golf experts,” which should be corrected. Statistical reporting needs to be more precise: exact p-values, effect sizes, and confidence intervals should be presented consistently. The mediation analysis is important but currently underdeveloped; including a dedicated table or figure summarizing the indirect and direct effects would make the findings clearer. Results should focus on the most significant outcomes, with less redundancy across text, figures, and tables.

The discussion restates the results at length but does not fully interpret their broader significance. It should more directly address the theoretical implications, such as the concept of neural efficiency and modular reorganization in motor learning. The narrative would benefit from linking the findings to practical applications, such as athlete training or rehabilitation. Citations are appropriate, but the discussion should be better integrated to show how this study advances beyond prior research, rather than only confirming it.

Reviewer #2: This text delves into how a long period of golf training has neuroplastic effects on the brain's structural covariance networks, providing unique clues about the brain changes related to a particular field. The work with personalized SCN is the main point of the method and it is a step further than the usual group-level studies. The relationship between variables as depicted by mediation analysis, linking training duration, clustering coefficient, and performance is very interesting.

Strengths:

Innovative utilization of the individual-specific SCN study.

Combining the global, nodal, and mediation outcomes allows for a more complete view.

Participant recruitment and ethical approval are both nicely organized.

The results are connected to the general research on motor training and neuroplasticity.

Weaknesses and points for improvement:

Causality limitation – Due to the cross-sectional design, the authors can't make definite statements about training-induced reconfiguration. They should soften the causal language and more strongly acknowledge this in the abstract and the discussion.

Sample size clarity – The abstract indicates that there were 30 per group, however, the methods report 20 in each group. The authors need to fix this inconsistency.

Data availability – The current limitations might not be enough to meet the journal requirements. The authors should think about releasing a fully anonymized version of their dataset or giving a clear enough reason that is acceptable to PLOS ONE.

Terminology consistency – In the Results section, there are mentions of “soccer experts,” which is probably an uncorrected artifact from previous templates. The authors should rectify this and use “golf experts” only.

Statistical reporting – Besides effect sizes, and, whenever possible, exact p-values should be reported to facilitate reproducibility.

Figures and captions – The figure captions not only should they detail the metrics but also their visualization for non-expert readers.

Limitations section – The authors, while noting the limitations of functional data, should also recognize that lifestyle, physical condition, or psychological traits might have influenced the differences between groups.

**Do you want your identity to be public for this peer review?** For information about this choice, including consent withdrawal, please see our Privacy Policy

Reviewer #1: No

Reviewer #2: No

---

## [Author Response · Author response to Decision Letter 1]

23 Oct 2025

Response to Reviewers

Reviewer 1

Q1. The abstract should be shortened and written more concisely to highlight the main findings and contributions without excessive technical detail.

A1. We have significantly shortened the abstract and removed redundant technical descriptions. The revised abstract now clearly emphasizes the core finding: that long-term golf training is associated with enhanced local network specialization (higher clustering coefficient and local efficiency) and that this reconfiguration partially mediates the link between training duration and performance.

Q2. The number of participants must be consistent across the abstract and the Methods section.

A2. We apologize for this inconsistency. Both the abstract and Methods section now correctly state that the study included 20 expert golfers, 20 novice golfers, and 20 non-golfer controls (total N = 60). The erroneous “n = 30” in Table 1 has been corrected to “n = 20” throughout.

Q3. The introduction should explicitly state the limitations of prior studies focusing on isolated brain regions and univariate analyses, and then highlight how a network-based approach provides a novel contribution.

A3. We have restructured the Introduction to include a dedicated paragraph (paragraph 3) that explicitly critiques the limitations of univariate, region-of-interest approaches and articulates how our individualized structural covariance network (SCN) method offers a systems-level perspective on motor expertise.

Q4. There is an inconsistency where “soccer experts” appears instead of “golf experts.”

A4. This was a typographical error from a previous draft. We have thoroughly proofread the manuscript and confirmed that all instances now correctly refer to “golf experts” or “golf players.”

Q5. Statistical reporting needs to be more precise: exact p-values, effect sizes, and confidence intervals should be presented consistently.

A5. We have revised all statistical reporting. In Tables 3 and 4, we now report exact p-values, Cohen’s d effect sizes, and in the mediation analysis, we provide the 95% confidence interval for the indirect effect.

Q6. The mediation analysis is important but currently underdeveloped; including a dedicated table or figure summarizing the indirect and direct effects would make the findings clearer.

A6. We have revised the caption of Figure 5 to clarify the mediated results.

Q7. The discussion restates the results at length but does not fully interpret their broader significance. It should more directly address the theoretical implications, such as the concept of neural efficiency and modular reorganization.

A7. We have substantially revised the Discussion to reduce result repetition and strengthen theoretical interpretation. We now explicitly frame our findings within the “neural efficiency” and “modular reorganization” theories, discussing how local specialization may support the automatization of complex motor sequences in golf.

Reviewer 2

Q1. Due to the cross-sectional design, the authors can't make definite statements about training-induced reconfiguration. They should soften the causal language.

A1. We fully agree. Throughout the manuscript, we have replaced causal language (e.g., “induces”) with correlational or associative terms. In the Abstract, Introduction, and Discussion, we now explicitly state that our cross-sectional design precludes causal inference and that pre-existing differences cannot be ruled out.

Q2. The abstract indicates that there were 30 per group, however, the methods report 20 in each group.

A2. This has been corrected. The abstract, Methods, and Table 1 now consistently report 20 participants per group.

Q3. Data availability – The current limitations might not be enough to meet the journal requirements.

A3. In compliance with PLOS ONE policy, we have deposited the fully anonymized structural MRI data and derived SCN matrices are available at: https://github.com/russellei/Long-Term-Intensive-Golf-Training-Induces-Reconfiguration-of-brain-structural-covariance-networks.

Q4. Terminology consistency – In the Results section, there are mentions of “soccer experts.”

A4. This error has been corrected. The manuscript now exclusively uses “golf experts.”

Q5. Figures and captions – The figure captions should detail the metrics and their visualization for non-expert readers.

A5. We have rewritten all figure captions. For example, the caption for Figure 3 now reads: “Comparison of global network topological properties… ‘aCp’ denotes the area under the curve of the clustering coefficient, which reflects the degree of local interconnectedness in the brain network.”

Q6. The authors should also recognize that lifestyle, physical condition, or psychological traits might have influenced the differences between groups.

A6. We have added a new paragraph in the Limitations section (Section 5) acknowledging that unmeasured factors such as lifestyle, fitness, stress, or personality traits may contribute to group differences, and we recommend that future studies control for these variables.

---

## [Decision Letter · Decision Letter 1]

20 Jan 2026

Dear Dr. Song,

Thank you for submitting your manuscript to PLOS ONE. After careful consideration, we feel that it has merit but does not fully meet PLOS ONE’s publication criteria as it currently stands. Therefore, we invite you to submit a revised version of the manuscript that addresses the points raised during the review process.

We look forward to receiving your revised manuscript.

Kind regards,

Cory James Coehoorn

Academic Editor

PLOS One

Journal Requirements:

Reviewers' comments:

Reviewer's Responses to Questions

**Comments to the Author**

Reviewer #1: All comments have been addressed

Reviewer #2: All comments have been addressed

2. Is the manuscript technically sound, and do the data support the conclusions?

Reviewer #1: Yes

Reviewer #2: Yes

3. Has the statistical analysis been performed appropriately and rigorously?

Reviewer #1: Yes

Reviewer #2: Yes

4. Have the authors made all data underlying the findings in their manuscript fully available?

Reviewer #1: No

Reviewer #2: Yes

5. Is the manuscript presented in an intelligible fashion and written in standard English?

Reviewer #1: Yes

Reviewer #2: Yes

Reviewer #1: Clarify statistical assumptions. Briefly specify whether the data met the assumptions for parametric testing and how correlation types were chosen.

Data availability. Consider depositing anonymized SCN matrices or summary statistics in a public repository, even under restricted access.

Minor linguistic refinements. The Discussion section could be slightly condensed to avoid repetition and improve flow.

Terminology consistency. Ensure uniform usage of abbreviations (e.g., Cp, Eloc, Eg) throughout figures and text.

Reviewer #2: This is a well-executed and articulate study that has made novel contributions to knowledge on experience-dependent reorganization of the brain network associated with motor skills mastery. I have found the manuscript free of remaining concerns and can be published as it is.

**Do you want your identity to be public for this peer review?** For information about this choice, including consent withdrawal, please see our Privacy Policy

Reviewer #1: No

Reviewer #2: No

---

## [Author Response · Author response to Decision Letter 2]

23 Jan 2026

Response to Reviewers

Editor and Reviewers,

Thank you for the opportunity to revise our manuscript titled “Long-Term Intensive Golf Training Induces Reconfiguration of Brain Structural Covariance Networks” (PONE-D-25-36556R1). We sincerely appreciate the reviewers’ insightful and constructive comments, which have significantly helped us improve the quality and clarity of the manuscript.

We have carefully addressed all the points raised in the review, particularly those by Reviewer #1, and have made corresponding revisions throughout the text. In the revised manuscript, all changes are red color.

Below, we detail our point-by-point responses to the reviewers’ comments.

Reviewer #1:

1. Clarify statistical assumptions. Briefly specify whether the data met the assumptions for parametric testing and how correlation types were chosen.

Response: We thank the reviewer for this suggestion. We have now clarified the statistical assumptions in the Statistical Analysis section (Section 2.5). Specifically, we have added that: The permutation tests used for group comparisons are non-parametric and do not assume normality, providing a robust and distribution-free assessment. Prior to correlation analyses, we assessed normality using the Shapiro–Wilk test. Variables conforming to normality were analyzed with Pearson’s correlation, otherwise Spearman’s correlation was used. In this study, all variables met normality assumptions, so Pearson’s correlation was applied.

2.Data availability. Consider depositing anonymized SCN matrices or summary statistics in a public repository, even under restricted access.

Response:We thank the reviewer for this valuable suggestion. In response to your comment, we have now publicly deposited the anonymized individual structural covariance network (SCN) matrices and all derived network metrics, along with the complete analysis code, in a GitHub repository. The repository is publicly accessible without restriction. We have also included a Data Availability Statement at the end of the manuscript that provides the direct link to this repository, ensuring full transparency and compliance with PLOS ONE's data policy.

3.Minor linguistic refinements. The Discussion section could be slightly condensed to avoid repetition and improve flow.

Response: We thank the reviewer for the suggestion. We have carefully revised the Discussion section to improve clarity, reduce redundancy, and enhance logical flow

4.Terminology consistency. Ensure uniform usage of abbreviations (e.g., Cp, Eloc, Eg) throughout figures and text.

Response: We have thoroughly checked the entire manuscript, figures, and tables to ensure consistent use of abbreviations for network metrics

Reviewer #2:

This is a well-executed and articulate study that has made novel contributions to knowledge on experience-dependent reorganization of the brain network associated with motor skills mastery. I have found the manuscript free of remaining concerns and can be published as it is.

Response: We sincerely thank the reviewer for the positive evaluation and endorsement of our work. We are pleased that the reviewer found the manuscript articulate and novel. In response to Reviewer #1’s comments, we have made several refinements to further improve clarity and rigor, which we believe strengthen the manuscript without altering its core contributions.

---

## [Decision Letter · Decision Letter 2]

11 Feb 2026

Dear Dr. Song,

Thank you for submitting your manuscript to PLOS ONE. After careful consideration, we feel that it has merit but does not fully meet PLOS ONE’s publication criteria as it currently stands. Therefore, we invite you to submit a revised version of the manuscript that addresses the points raised during the review process.

We look forward to receiving your revised manuscript.

Kind regards,

Cory James Coehoorn

Academic Editor

PLOS One

Journal Requirements:

Reviewers' comments:

Reviewer's Responses to Questions

**Comments to the Author**

Reviewer #1: All comments have been addressed

2. Is the manuscript technically sound, and do the data support the conclusions?

Reviewer #1: Yes

3. Has the statistical analysis been performed appropriately and rigorously?

Reviewer #1: Yes

4. Have the authors made all data underlying the findings in their manuscript fully available?

Reviewer #1: Yes

5. Is the manuscript presented in an intelligible fashion and written in standard English?

Reviewer #1: Yes

Reviewer #1: 1. These are typos

2. Participant Number Discrepancy:In the Abstract (according to the file snippet/metadata), it states: "A total of 60 participants—30 expert golfers, 30 novice golfers, and 30 non-golfer controls." (Note: 30+30+30=90, not 60).In Section 2.1 (Subjects), it states: "The Pro group comprised 20... The New group included 20... The control group consisted of 20." (Total 60). Ensure the numbers in the Abstract match the Methods exactly (likely 20 per group).

3. Contradiction in Mediation Analysis (Partial vs. Full). You state the clustering coefficient "partially mediated the association."Results (Section 3.3): You state: "The direct effect of training duration on stroke performance became non-significant after controlling for Cp. Check your statistical output. If the direct path is indeed non-significant (p > .05), change the Abstract to say "fully mediated" or simply "mediated". If it is significant but reduced, change the Results text to reflect that.

4. Anatomical Terminology Error ("Crus 10"): You refer to "right cerebellar Crus 10" (or Crus X). Results (3.2): You refer to "right cerebellum-10". Change "Crus 10" in the Discussion to "Lobule X" or "Cerebellum Region 10" to be anatomically correct.

5. Ensure you explicitly argue why reduced integration in the visual cortex helps. (e.g., "The reduction in global efficiency may reflect a pruning of irrelevant connections to prevent visual distractors, focusing resources on local processing...")

**Do you want your identity to be public for this peer review?** For information about this choice, including consent withdrawal, please see our Privacy Policy

Reviewer #1: No

---

## [Author Response · Author response to Decision Letter 3]

11 Feb 2026

Reviewer #1:

2.Participant Number Discrepancy:In the Abstract (according to the file snippet/metadata), it states: "A total of 60 participants—30 expert golfers, 30 novice golfers, and 30 non-golfer controls." (Note: 30+30+30=90, not 60).In Section 2.1 (Subjects), it states: "The Pro group comprised 20... The New group included 20... The control group consisted of 20." (Total 60). Ensure the numbers in the Abstract match the Methods exactly (likely 20 per group).

Re: Thank you for catching this discrepancy. The correct sample size is 20 per group, totaling 60 participants. We have revised the Abstract accordingly to state “20 expert golfers, 20 novice golfers, and 20 non-golfer controls” and removed the inconsistent numbers. The Methods section remains unchanged.

3.Contradiction in Mediation Analysis (Partial vs. Full). You state the clustering coefficient "partially mediated the association."Results (Section 3.3): You state: "The direct effect of training duration on stroke performance became non-significant after controlling for Cp. Check your statistical output. If the direct path is indeed non-significant (p > .05), change the Abstract to say "fully mediated" or simply "mediated". If it is significant but reduced, change the Results text to reflect that.

Re: Thank you for your careful reading and for pointing out the imprecise wording regarding the mediation result. We have revised the abstract accordingly.

In the Results section (3.3), we reported that the direct effect of training duration on stroke performance became non-significant (β = −0.22, p = 0.15) after including the clustering coefficient (Cp) as a mediator, while the indirect effect was significant (β = −0.1, 95% CI [−0.37, −0.09]). According to standard terminology, this pattern indicates full mediation, not partial mediation.

We have now corrected the abstract to state that the clustering coefficient “mediated” the association, removing the word “partially” to accurately reflect the statistical finding. The revised sentence now reads:

“Importantly, the clustering coefficient mediated the association between training duration and stroke accuracy.”

4.Anatomical Terminology Error ("Crus 10"): You refer to "right cerebellar Crus 10" (or Crus X). Results (3.2): You refer to "right cerebellum-10". Change "Crus 10" in the Discussion to "Lobule X" or "Cerebellum Region 10" to be anatomically correct.

Re Thank you for pointing out the anatomical error. We have corrected “right cerebellum-10” and “Crus 10” to “right Cerebellum Region 10” throughout the manuscript, tables, and figures to align with AAL atlas conventions.

5.Ensure you explicitly argue why reduced integration in the visual cortex helps. (e.g., "The reduction in global efficiency may reflect a pruning of irrelevant connections to prevent visual distractors, focusing resources on local processing...")

Re Thank you for your comment. We agree that merely reporting reduced global efficiency in the left calcarine cortex is insufficient; we need to explicitly argue why this reduction is functionally beneficial.

We have revised the Discussion accordingly. The new paragraph interprets this decrease as a functional pruning process, whereby long-term training suppresses task-irrelevant visual connections to enhance focal attention and reduce perceptual noise—critical demands in golf. We now also cite evidence from gymnasts and shooters showing similar visual cortex reductions alongside superior performance, supporting this as an adaptive neural efficiency effect rather than degradation. The revised text is included in Section 4.

---

## [Decision Letter · Decision Letter 3]

18 Feb 2026

Long-Term Intensive Golf Training Induces Reconfiguration of brain structural covariance networks

PONE-D-25-36556R3

Dear Dr. Song,

We’re pleased to inform you that your manuscript has been judged scientifically suitable for publication and will be formally accepted for publication once it meets all outstanding technical requirements.

Kind regards,

Cory James Coehoorn

Academic Editor

PLOS One

Additional Editor Comments (optional):

Reviewers' comments:

Reviewer's Responses to Questions

**Comments to the Author**

Reviewer #1: All comments have been addressed

2. Is the manuscript technically sound, and do the data support the conclusions?

Reviewer #1: Yes

3. Has the statistical analysis been performed appropriately and rigorously?

Reviewer #1: Yes

4. Have the authors made all data underlying the findings in their manuscript fully available?

Reviewer #1: Yes

5. Is the manuscript presented in an intelligible fashion and written in standard English?

Reviewer #1: Yes

Reviewer #1: (No Response)

**Do you want your identity to be public for this peer review?** For information about this choice, including consent withdrawal, please see our Privacy Policy

Reviewer #1: No

---

## [Editor Report · Acceptance letter]

PONE-D-25-36556R3

PLOS One

Dear Dr. Song,

I'm pleased to inform you that your manuscript has been deemed suitable for publication in PLOS One. Congratulations! Your manuscript is now being handed over to our production team.

Kind regards,

on behalf of

Dr. Cory James Coehoorn

Academic Editor

PLOS One